Predicting Pinus monophylla forest cover in the Baja California Desert by remote sensing

http://orcid.org/0000-0003-4265-9420 Escobar-Flores Jonathan G. 1
http://orcid.org/0000-0002-5135-5739 Lopez-Sanchez Carlos A. 2
Sandoval Sarahi 3
Marquez-Linares Marco A. 1
Wehenkel Christian 2 wehenkel@ujed.mx
1 Centro Interdisciplinario De Investigación para el Desarrollo Integral Regional, Unidad Durango, Instituto Politécnico Nacional , Durango, Durango , México
2 Instituto de Silvicultura e Industria de la Madera, Universidad Juárez del Estado de Durango , Durango , Mexico
3 CONACYT—Instituto Politécnico Nacional, CIIDIR Unidad Durango , Durango, Durango , México
Winkler Robert
Electronic publication date: 2018 Apr 4
Publication date: 2018
Volume: 6
Electronic Location ID: e4603
Received 2017 Nov 29; Accepted 2018 Mar 21
Copyright: © 2018 Escobar-Flores et al.
Copyright year: 2018
Copyright holder: Escobar-Flores et al.
License: This is an open access article distributed under the terms of the Creative Commons Attribution License, which permits unrestricted use, distribution, reproduction and adaptation in any medium and for any purpose provided that it is properly attributed. For attribution, the original author(s), title, publication source (PeerJ) and either DOI or URL of the article must be cited.
License URL: https://creativecommons.org/licenses/by/4.0/

Keywords: DEM, Sentinel-2, Ruggedness, Remote sensing, Neural net, Forest, Baja California, NDVI, Kappa, Classification

Funding: National Council of Science and Technology (CONACYT) #205965 This work was supported by the National Council of Science and Technology (CONACYT) by grant #205965. The funders had no role in study design, data collection and analysis, decision to publish, or preparation of the manuscript.

==============================
The Californian single-leaf pinyon (Pinus monophylla var. californiarum), a subspecies of the single-leaf pinyon (the world’s only one-needled pine), inhabits semi-arid zones of the Mojave Desert (southern Nevada and southeastern California, US) and also of northern Baja California (Mexico). This tree is distributed as a relict subspecies, at elevations of between 1,010 and 1,631 m in the geographically isolated arid Sierra La Asamblea, an area characterized by mean annual precipitation levels of between 184 and 288 mm. The aim of this research was (i) to estimate the distribution of P. monophylla var. californiarum in Sierra La Asamblea by using Sentinel-2 images, and (ii) to test and describe the relationship between the distribution of P. monophylla and five topographic and 18 climate variables. We hypothesized that (i) Sentinel-2 images can be used to predict the P. monophylla distribution in the study site due to the finer resolution (×3) and greater number of bands (×2) relative to Landsat-8 data, which is publically available free of charge and has been demonstrated to be useful for estimating forest cover, and (ii) the topographical variables aspect, ruggedness and slope are particularly important because they represent important microhabitat factors that can determine the sites where conifers can become established and persist. An atmospherically corrected a 12-bit Sentinel-2A MSI image with 10 spectral bands in the visible, near infrared, and short-wave infrared light region was used in combination with the normalized differential vegetation index (NDVI). Supervised classification of this image was carried out using a backpropagation-type artificial neural network algorithm. Stepwise multiple linear binominal logistical regression and Random Forest classification including cross validation were used to model the associations between presence/absence of P. monophylla and the five topographical and 18 climate variables. Using supervised classification of Sentinel-2 satellite images, we estimated that P. monophylla covers 6,653 ± 319 ha in the isolated Sierra La Asamblea. The NDVI was one of the variables that contributed most to the prediction and clearly separated the forest cover (NDVI > 0.35) from the other vegetation cover (NDVI < 0.20). Ruggedness was the most influential environmental predictor variable, indicating that the probability of occurrence of P. monophylla was greater than 50% when the degree of ruggedness terrain ruggedness index was greater than 17.5 m. The probability of occurrence of the species decreased when the mean temperature in the warmest month increased from 23.5 to 25.2 °C. Ruggedness is known to create microclimates and provides shade that minimizes evapotranspiration from pines in desert environments. Identification of the P. monophylla stands in Sierra La Asamblea as the most southern populations represents an opportunity for research on climatic tolerance and community responses to climate variability and change.

Introduction

The Californian single-leaf pinyon (Pinus monophylla var. californiarum), a subspecies of the single-leaf pinyon (the world’s only one-needled pine), inhabits semi-arid zones of the Mojave Desert (southern Nevada and southeastern California, US) and also of northern Baja California (BC) (Mexico). It is both cold-tolerant and drought-resistant and is mainly differentiated from the typical subspecies P. monophylla var. monophylla by a larger number of leaf resin canals and longer fascicle-sheath scales (Bailey, 1987). This subspecies was first reported in BC in 1767 (Bullock & Heath, 2006). The southernmost record of P. monophylla var. californiarum in America was previously in BC, 26–30 miles north of Punta Prieta, at an elevation of 1,280 m (longitude −114°.155; latitude 29°.070, catalogue number ASU 0000235), and the type specimen is held in the Arizona State University Vascular Plant Herbarium.

This species is distributed as a relict subspecies in the geographically isolated Sierra La Asamblea, at a distance of 196 km from the Southern end of the Sierra San Pedro Martir and at elevations of between 1,010 and 1,631 m (Moran, 1983) in areas with mean annual precipitation levels of between 184 and 288 mm (Roberts & Ezcurra, 2012). The Californian single-leaf pinyon grows together with up to about 86 endemic plant species, although the number of species decreases from north to south (Bullock et al., 2008).

Adaptation of P. monophylla var. californiarum to arid ecosystems enables the species to survive annual precipitation levels of less than 150 mm. In fact, seeds of this variety survive well under shrubs such as Quercus spp. and Arctostaphylus spp., a strategy that enables the pines to widen their distribution, as has occurred in the great basin in California (Callaway et al., 1996; Chambers, 2001), and for them to occupy desert zones such as Sierra de la Asamblea. Despite the importance of this relict pine species, its existence is not considered in most forest inventories in Mexico (CONABIO, 2017).

Remote sensing with Landsat images has been demonstrated to be useful for estimating forest cover; the Landsat-8 satellite has sensors (seven bands) that can be predicted vegetation attributes at a spatial resolution of 30 m (Madonsela et al., 2017). However, the European Space Agency’s Copernicus program has made Sentinel-2 satellite images available to the public free of charge. The spatial resolution (10 m is pixel) of the images is three times finer that of Landsat images, thus increasing their potential for predicting and differentiating types of vegetation cover (Drusch et al., 2012; Borràs et al., 2017). The Sentinel-2 has 13 bands, of which 10 provide greater-quality radiometric images of spatial resolution 10–20 m in the visible and infrared regions of the electromagnetic spectrum. These images are therefore ideal for land classification (ESA, 2017).

The aim of this research was (i) to estimate the distribution of P. monophylla var. californiarum in Sierra La Asamblea, Baja California (Mexico) by using Sentinel-2 images, and (ii) to test and describe the relationship between this distribution of P. monophylla and five topographic and 18 climate variables. We hypothesized that (i) the Sentinel-2 images can be used to accurately predict the P. monophylla distribution in the study site due to finer resolution (×3) and greater number of bands (×2) than in Landsat-8 data, and (ii) the topographical variables aspect, ruggedness and slope are particularly influential because they represent important microhabitat factors that can determine where conifers can become established and persist (Marston, 2010).

Materials and Methods

Study area

Sierra La Asamblea is located in Baja California’s central desert (−114°9′W 29°19′N, elevation range 280–1,662 m, Fig. 1). The climate in the area is arid, with maximum temperatures of 40 °C in the summer (García, 1998). The sierra is steeper on the western slopes, with an average incline of 35°, and with numerous canyons with occasional springs and oases. Valleys and plateaus are common in the proximity of the Gulf of California. Granite rocks occur south of the sierra and meta-sedimentary rocks along the north and southeast of the slopes. The predominant type of vegetation is xerophilous scrub, which is distributed at elevations ranging from 200 to 1,000 m. Chaparral begins at an altitude of 800 m, and representative specimens of Adenostoma fasciculatum, Ambrosia ambrosioides, Dalea bicolor orcuttiana, Quercus tuberculata, Juniperus california, and P. monophylla are also present at elevations above 1,000 m. Populations of the endemic palm tree Brahea armata also occur in the lower parts of the canyons with superficial water flow and through the rocky granite slopes (Bullock & Heath, 2006).

Figure 1 Map of Sierra La Asamblea.

The black circles indicate georeferenced sites occupied by Pinus monophylla.

Datasets

Sentinel-2

The Sentinel-2A multispectral instrument (MSI) L1C dataset, acquired on October 11, 2016, in the trajectory of coordinates latitude 29°.814, longitude 114°.93, was downloaded from the US. Geological Survey (USGS) Global Visualization Viewer at http://glovis.usgs.gov/. The 12-bit Sentinel-2A MSI image has 13 spectral bands in the visible, NIR, and SWIR wavelength regions with spatial resolutions of 10–60 m. However, band one, used for studies of coastal aerosols, and bands nine and 10, applied for, respectively, water vapor correction and cirrus detection, were not used in this study (ESA, 2017). Hence, the data preparation involved four bands at 10 m and the resampling of the six S2 bands acquired at 20 m to obtain a layer stack of 10 spectral bands at 10 m (Table 1) using the ESA’s Sentinel-toolbox ESA Sentinel Application Platform (SNAP) and then converted to ENVI format.

Table 1 Sentinel-2 spectral bands used to predict the Pinus monophylla forest cover.

Band	Central wavelength (μm)	Resolution (m)	
Band 2—Blue	0.490	10	
Band 3—Green	0.560	10	
Band 4—Red	0.665	10	
Band 5—Vegetation red edge	0.705	20	
Band 6—Vegetation red edge	0.740	20	
Band 7—Vegetation red edge	0.783	20	
Band 8—NIR	0.842	10	
Band 8A—Vegetation red edge	0.865	20	
Band 11—SWIR	1.610	20	
Band 12—SWIR	2.190	20	

Because atmospherically improved images are essential to enable assessment of spectral indices with spatial reliability and product comparison, Level-1C data were converted to Level-2A (bottom of atmosphere reflectance) by taking into account the effects of aerosols and water vapor on reflectance (Radoux et al., 2016). The corrections were made using the Sen2Cor tool (Telespazio VEGA Deutschland GmbH, 2016) for Sentinel-2 images.

The following equation was used to calculate the normalized difference vegetation index (NDVI): NDVI = (NIR − R)/(NIR + R), where NIR is the near infrared light (band) reflected by the vegetation, and R is the visible red light reflected by the vegetation (Rouse et al., 1974). The NDVI is useful for discriminating the layers of temperate forest from scrub and chaparral. Areas occupied by large amounts of unstressed green vegetation will have values much greater than 0 and areas with no vegetation will have values close to 0 and, in some cases, negative values (Pettorelli, 2013). The NDVI image was combined with the previously described multi spectral bands.

Environmental variables

Tree species distribution is generally modulated by hydroclimate and topographical variables (Elliott et al., 2005; DeCastilho et al., 2006), which can be estimated from digital terrain models (DTM) (Osem et al., 2009; Spasojevic et al., 2016). A DTM was obtained by using tools available from the Instituto Nacional de Estadistica y Geografía (http://www.inegi.org.mx/geo/contenidos/datosrelieve) with a spatial resolution of 15 m. The DTM was processed with the QGIS (QGIS Development Team, 2016), using Terrain analysis tools, elevation, slope and aspect (Table 2).

Table 2 Topographical and climatic variables considered in the study.

Variable	Abbreviation	Units	Mean	SD	Max	Min	
Terrain ruggedness index	TRI	m	20.33	6.66	35.90	4.69	
Vector ruggedness measure	VRM	NA	0.005	0.007	0.13	0	
Slope	S	°	28.38	8.92	48.34	3.42	
Aspect*	A	°	190.51	68.72	350.44	20.55	
Elevation*	E	m	1,302.41	124.96	1,631	1,010	
Mean annual temperature*	MAT	°C	16.57	0.38	17.4	15.5	
Mean annual precipitation*	MAP	mm	229.56	19.95	288	184	
Growing season precipitation, April–September*	GSP	mm	79.08	9.60	108	57	
Mean temperature in the coldest month*	MTCM	°C	10.85	0.37	11.7	9.8	
Minimum temperature in the coldest month*	MMIN	°C	3.42	0.41	4.3	2.3	
Mean temperature in the warmest month	MTWM	°C	24.52	0.31	25.2	23.5	
Maximum temperature in the warmest month	MMAX	°C	34.10	0.31	34.7	33.1	
Julian date of the last freezing data of spring*	SDAY	Days	82.57	7.86	106	60	
Julian date of the first freezing data of autumn*	FDAY	Days	331.28	2.62	339	324	
Length of the frost-free period*	FFP	Days	259.22	8.36	285	240	
Degree days ˃5 °C*	DD5	Days	4,245.26	137.52	4,550	3,852	
Degree days ˃5 °C accumulating within the frost-free period*	GSDD5	Days	3,491.82	164.76	3,944	2,995	
Julian date when the sum degree days ˃5 °C reaches 100*	D100	Days	17.07	1.10	20	15	
Degree days ˂0 °C*	DD0	Days	0	0	0	0	
Minimum degree days ˂0 °C*	MMINDD0	Days	8.07	20.29	145	45	
Spring precipitation	SPRP	mm	7.54	0.71	10	6	
Summer precipitation*	SMRP	mm	43.74	6.29	62	29	
Winter precipitation*	WINP	mm	110.93	7.93	133	93	
Note:

* Variables for which no significant difference between the medians was obtained after Bonferroni correction (α = 0.0005) were excluded from further analysis.

The ruggedness was estimated using two indexes: (i) the terrain ruggedness index (TRI) of Riley, Degloria & Elliot (1999) and (ii) a vector ruggedness measure (VRM), both implemented in QGIS (QGIS Development Team, 2016). The TRI computes the values for each grid cell of a DEM. This calculates the sum change in elevation between a grid cell and its eight-neighbor grid cell. VRM incorporates the heterogeneity of both slope and aspect. This measure of ruggedness uses three-dimensional dispersion of vectors normal to planar facets on landscape. This index lacks units and ranges from 0 (indicating a totally flat area) to 1 (indicating maximum ruggedness) (Sappington, Longshore & Thompson, 2007).

In addition, 18 climate variables with a 30-arc second resolution (approximate 800 m) (Table 2) were obtained from a national database managed by the University of Idaho (http://charcoal.cnre.vt.edu/climate) and which requires point coordinates (latitude, longitude and elevation) as the main inputs (Rehfeldt, 2006; Rehfeldt et al., 2006). These variables are frequently used to study the potential effects of global warming on forests and plants in Western North America and Mexico (Sáenz-Romero et al., 2010; Silva-Flores, Pérez-Verdín & Wehenkel, 2014).

Pixel-based classification

Classification method

Pixel-based classification was carried out in order to predict four different types of land cover in the study area (P. monophylla, scrub, chaparral, and no apparent vegetation). A supervised classification approach with a backpropagation-type artificial neural network (BPNN) (Tan & Smeins, 1996) was applied. BPNN is widely used because of its structural simplicity and robustness in modeling non-linear relationships. In this study, the BPNN comprises a set of three layers (raster): an input layer, a hidden layer, and an output layer (Richards, 1999). Each layer consists of a series of parallel processing elements (neurons or nodes). Each node in a layer is linked to all nodes in the next layer (Guo et al., 2013).

The first step in BPNN supervised classification is to enter the input layer, which in this study corresponded to the values of the pixels of 10 Sentinel-2 bands and of the NDVI image. Weights were then assigned to the BPNN to produce analytical predictions from the input values. These data were contrasted with the category to which each training pixel belongs, corresponding to Georeferenced sites (Datum WGS-84, 11N) obtained in the field in October 2014 and October 2015.

A stratified random sampling method (Olofsson et al., 2013) was used to generate the reference data in QGIS software (QGIS Development Team, 2016). A total of 2,143 random points were sampled, with at least 400 points for each class (Goodchild, 1994). The following classes were considered: (i) P. monophylla, 536 sites, (ii) scrub, 764 sites, (iii) chaparral, 405 sites, and (iv) no apparent vegetation, 438 sites (INEGI, 2013). Class discrimination processes occurred in the hidden layer and the synapses between the layers were estimated by an activation function. We used a logistic function and training rate of 0.20, previously applied to land cover classification (Hepner et al., 1990; Richards, 1999; Braspenning & Thuijsman, 1995). Learning occurs by adjusting the weights in the node to minimize the difference between the output node activation, and BPNN then calculates the error at each iteration with root mean square (RMS) error. The output layer comprised four neurons representing the four target classes of land cover (P. monophylla, Scrub, Chaparral, and no apparent vegetation). Average spectral signatures for the four different types of land cover are shown in Fig. 2.

Figure 2 Average spectral signatures of cover vegetation in Sierra La Asamblea, Baja California.

Validation

The BPNN classification was cross-validated (10-fold) using a confusion matrix, which is a table that compares the reference data and the classification results. We estimated the uncertainty of the classification using estimated error matrix in terms of proportion of area and estimates of overall map accuracy (O^), user’s accuracy (U^i) (or commission error), and producer’s accuracy (P^j) (or omission error) recommended by Olofsson et al. (2013): pij is defined as a cell entry of error matrix of i map classes. A poststratified estimator of pij is: (1) p^ij=Winijni.

where Wi is the proportion of the area mapped as class i. ni. is the total number of sample units in map class i. nij is the sample count at cell (i, j) in the error matrix.

p^.j is a poststratified estimator for simple random and systematic sampling: (2) p^.j=∑i=1qWinijni.

where q is the class number.

An unbiased estimator of the total area of class j is then (3) A^j=A⋅p^.j

where A is the total map area. For p^.j, the standard error is estimated by (Cochran, 1977): (4) S(p^.j)=∑i=1qWi2nijni.(1−nijni.)ni.−1

The standard error of the error-adjusted estimated area is (5) S(A^j)=A⋅S(p^.j)

Finally, (6) A^j±1.96⋅S(A^j)

presents an approximate 95% confidence interval.

The O^, U^i, and P^j were calculated with Eqs. (7)–(9) (Congalton, 1991). U^i of class i is the proportion of the area mapped as class i that has reference class i. P^j of class j is the proportion of the area of reference class j that is mapped as class j.

(7) O^=∑j=1qp^jj

(8) U^i=p^iip^i.

(9) P^j=p^jjp^.j

We then generated a map from the results of the probability of class assignment. The accuracy of classification was also estimated using the Kappa (K) coefficient. The K coefficient is often used as an overall measure of accuracy (Abraira, 2001). This coefficient takes values of between 0 and 1, where values close to 1 indicate a greater degree of agreement between classes and observations, and a value of 0 suggests that the observed agreement is random. However, the use of K is controversial because (i) K would underestimate the probability that a randomly selected pixel is correctly classified, (ii) K is greater correlated with overall accuracy so reporting Kappa is redundant for overall accuracy (Olofsson et al., 2014).

Relationship between presence of P. monophylla and environmental variables

To model and test the association between presence/absence of P. monophylla in the study area and topographical or climate variables, a Kruskal–Wallis test was used to estimate the difference in the median values in relation to presence and absence of P. monophylla. All variables for which no significant difference between the median values was predicted after Bonferroni correction (α = 0.0005) were excluded from further analysis. The collinearity between the variables with a significant difference between the medians of presence and absence was estimated using the Spearman correlation coefficient (rs). When the rs value for the difference between two variables was greater than 0.7, only the variable with the lowest p value in the Kruskal–Wallis test was used in the models (as reported by Salas, Valdez & Michel, 2017 and Shirk et al., 2018). Finally, stepwise multiple linear binominal logistical regression and Random Forest classification including cross valuation (10-fold) were used to model the associations between presence/absence of P. monophylla and the most important topographical and climate variables (Shirk et al., 2018).

Regression and classification including cross-validations were carried out using the trainControl, train, glm (family = “binomial”), and rf functions, as well as the “randomForest” and “caret” packages (Venables & Ripley, 2002) in R (version 3.3.2) (R Development Core Team, 2017). The goodness-of-fit of the logistical regression model was evaluated using the Akaike information criterion, root mean square error, and residual deviance. Validation of the randomForest model was performed using under the curve (AUC; Fawcett, 2006), true skill statistic (TSS; Allouche, Tsoar & Kadmon, 2006), Kappa (Abraira, 2001), specificity, and sensitivity.

Results

Pixel-based classification

We estimated the area of P. monophylla cover with a margin of error (at approximate 95% confidence interval) of 6,653 ± 319 (standard error) ha in Sierra de la Asamblea, Baja California, Mexico (Fig. 3). The confusion matrix of the accuracy assessment is listed in Table 3 including user’s and producer’s accuracy for each class. The supervised classification with BPNN yielded predictions with an overall accuracy of identification of 87.74% (Table 4). This level of accuracy was estimated in the 32 interactions with 0.04 RMS training. The proportion of omission errors in the P. monophylla class was only 2.62%, i.e., 97.39% of the pixels were correctly classified. The shrub class had the larger proportion of omission errors (18.98%). The value of NDVI in the P. monophylla forest fluctuated between 0.30 and 0.41, and in chaparral between 0.24 and 0.28. The smallest values of NDVI corresponded to scrub vegetation, with values between 0.10 and 0.15.

Figure 3 (A) Estimated land cover classes using BPNN classification in Sierra La Asamblea. (B) Probability map of class assignment.

Table 3 Estimated error matrix based of sample counts (nij) from the accuracy assessment sample.

	Classes	Reference	Total	Map area (ha)	Wi	
P	S	C	WV	
Map	P	522	0	14	0	536	5,395	0.169	
S	24	619	119	2	764	12,309	0.387	
C	50	0	348	7	405	8,206	0.258	
NAP	0	0	20	418	438	5,913	0.186	
Total	596	619	501	427	2,143	31,823	1	
Notes:

Map classes are the rows while the reference classes are the columns.

P, Pinus monophylla; S, shrub; C, chaparral; NAP, no apparent vegetation; Wi, proportion of the area mapped as class i.

Table 4 Error matrix of four classes with cell entries (pij) based on Table 3 and expressed in terms of proportion of area.

	Classes	References	Accuracy	
P	S	C	WV	User’s	Producer’s	Overall	
Map	P	0.1651	0.0000	0.0044	0.0000	0.974 ± 0.07	0.790 ± 0.04	0.877 ± 0.01	
S	0.0122	0.3134	0.0602	0.0010	0.810 ± 0.02	1.000		
C	0.0318	0.0000	0.2216	0.0045	0.859 ± 0.01	0.752 ± 0.07		
NAP	0.0000	0.0000	0.0085	0.1773	0.954 ± 0.002	0.970 ± 0.02		
Total	0.2091	0.3134	0.2947	0.1828				
Notes:

Accuracy measures are presented with a 95% confidence interval. Map classes (rows), reference classes (columns).

P, Pinus monophylla; S, shrub; C, chaparral; NAP, no apparent vegetation.

Relationship between presence of P. monophylla and environmental variables

The Kruskal–Wallis test indicated that the median values for ruggedness TRI (p < 2.1 × 10−16), slope (p < 2.2 × 10−16), ruggedness VRM (p = 4.9 × 10−9), mean temperature in the warmest month (MTWM) (p = 0.000014), Maximum temperature in the warmest month (MMAX) (p = 0.000048), and SPRP (p = 0.00037) were most variable between sites in which P. monophylla was present and absent. The variable slope was closely correlated with ruggedness as well as with MMAX and MTWM (rs > 0.7). The pslope of the Kruskal–Wallis test was larger than pruggedness and pMMAX was larger than pMTWM. Slope and MMAX were therefore excluded from the model analysis. The stepwise multiple linear binominal logistical and Random Forest models showed that the “presence of P. monophylla” model included the independent variables ruggedness, ruggedness VRM, and average temperature in the warmest month (MTWM) (Table 5).

Table 5 Results of the multiple linear binomial logistic regression model (AIC = 601.8; residual deviance = 593.85 on 588 degrees of freedom), TRI, terrain ruggedness index; VRM, vector ruggedness measure; MTWM, mean temperature in the warmest month.

Variable	Estimate	Std. Error	Z value	Pr(>|z|)	
Intercept	25.351	8.895	2.850	0.0044	
MTWM	−1.159	0.362	−3.201	0.0014	
TRI	0.178	0.015	11.200	<2e−16	
VRM	28.476	13.847	2.056	0.0397	

The ruggedness factor was the most influential predictor variable and indicated that the probability of P. monophylla occurrence was larger than 50% when the degree of ruggedness TRI was greater than 17.5 m (Table 5). The ruggedness VRM also indicated that a minimum change in roughness increases the probability of presence of the pine. The probability of occurrence of P. monophylla decreased when MTWM increased from 23.5 to 25.2 °C (Table 5). After cross validation (10-fold), the Random Forest model revealed that the variables ruggedness TRI, ruggedness VRM, and MTWM yielded a greater correlation for their ability to predict presence of the P. monophylla (AUC = 0.920, TSS = 0.690, Kappa = 0.691). The sensitivity was 0.812 and specificity was 0.878.

Discussion

Pixel-based classification

Predicting the presence of pine forest by using BPNN proved feasible. The NDVI was one of the variables that contributed to the prediction and clearly separated forest cover (NDVI > 0.35) from the other types of vegetation cover (NDVI < 0.20). The overall accuracy of classification (K = 0.87) was similar to that reported in other studies using Sentinel-2A MSI images. For example, Immitzer, Vuolo & Atzberger (2016) reported a K of 0.85 for tree prediction in Europe by using five classes and a random forest classifier. Vieira et al. (2003) reported a K = 0.77 in eastern Amazon using seven classes and 1999 Landsat 7 ETM imagery. However, Sothe et al. (2017) reported K values of 0.98 and 0.90 for, respectively, three successional forest stages and field in a subtropical forest in Southern Brazil by using Sentinel-2 and Landsat-8 data associated with the support vector machine algorithm. Kun et al. (2014) estimated K values of 0.70 to 0.85 for land-use type prediction (including forest) in China by using the support vector machine algorithm classifier and Landsat-8 images of rougher spatial resolution than Sentinel images. The very greater accuracy of predictions by Kun et al. (2014) was probably due to the large-scale of the study and the clearly differentiated types of land considered.

Relationship between presence of P. monophylla and environmental variables

Ruggedness of the terrain was the most important topographic variable, significantly explaining the presence of pines in Sierra La Asamblea (Table 5). Ruggedness, which is strongly positively correlated with slope, may reduce solar radiation, air temperature, and evapotranspiration due to increased shading (Tsujino et al., 2006; Bullock et al., 2008; Allen et al., 2010). The ruggedness indicated by the TRI index explains the presence of the pines because Sierra La Asamblea is heterogeneous in terms of elevation. The VRM index was less important partly because the index is strongly dependent on the vector aspect (Gisbert & Martí, 2010) and in the case of Sierra Asamblea the aspect is very homogeneous and the index values therefore tend to be very low (Table 5), as also reported by Wu et al. (2018). The pines were expected to colonize north facing slopes, which are exposed to less solar radiation than slopes facing other directions. However, the topographical variable aspect was not important in determining the presence of P. monophylla var. californiarum in the study site, possibly because of physiological adaptations regarding water-use efficiency and photosynthetic nitrogen-use efficiency (DeLucia & Schlesinger, 1991), as reported for the P. monophylla, P. halepensis, P. edulis, and P. remota in arid zones (Lanner & Van Devender, 2000; Helman et al., 2017). The Mediterranean climate, with wet winters and dry summers, is another characteristic factor in this mountain range. In the winter in this part of the northern hemisphere, the sun (which is in a lower position and usually affects the southern aspect by radiation) is masked by clouds, rainfall and occasional snowfall (León-Portilla, 1988). During the summer, the solar radiation is more intense, but similar in all directions because the sun is closest to its highest point (Stage & Salas, 2007).

The above-mentioned finding contrasts with those of other studies reporting that north-eastern facing slopes in the northern hemisphere receive less direct solar radiation, thus providing more favorable microclimatic conditions (air temperature, soil temperature, soil moisture) for forest development, permanence, and productivity than southwest-facing sites (Åström et al., 2007; Stage & Salas, 2007; Huang et al., 2009; Marston, 2010; Klein et al., 2014). DeLucia & Schlesinger (1991) reported that P. monophylla populations in the Great Basin California desert with summer rainfall (monsoon) preferred an east-southeast aspect with less intense solar radiation and evapotranspiration.

The probability of occurrence of P. monophylla was also related to the climatic variable MTWM. In Sierra La Asamblea, this pine species was found in a narrow range of MTWM of between 23.5° and 25.2° (Table 1), which, however, is a smaller range than reported for the other pine species (Tapias et al., 2004; Roberts & Ezcurra, 2012). Therefore, this species should adapt well to greater temperatures in the summer (Lanner & Van Devender, 2000), which is usually a very dry period in the study site (León-Portilla, 1988). However, the probability of occurrence was greatest for an MTWM of 23.5 °C (Table 5), which occurred at the top of Sierra La Asamblea, at an elevation of about 1,660 m). We therefore conclude that this species can also grow well when the MTWM is below 23.5 °C. On the other hand, considering MTWM as factor yielded a probability of occurrence of 25%–80%. The spatial resolution of the climatic data by the national database run by the University of Idaho is probably not adequate for describing the microhabitat of P. monophylla (Rehfeldt et al., 2006; Marston, 2010).

Identification of the P. monophylla stands in Sierra La Asamblea as the most southern populations represents an opportunity for research on climatic tolerance and community responses to climatic variation and change.

Supplemental Information

Supplemental Information 1 Pine presence records.

Click here for additional data file.

Supplemental Information 2 Pine absence records.

Click here for additional data file.

We are grateful to E. Espinoza, F. Macias, and A. Guerrero for support with the fieldwork.

Additional Information and Declarations

Competing Interests

Author Contributions

Data Availability

The authors declare that they have no competing interests.

Jonathan G. Escobar-Flores conceived and designed the experiments, performed the experiments, analyzed the data, contributed reagents/materials/analysis tools, prepared figures and/or tables, authored or reviewed drafts of the paper, approved the final draft, calculation of data and review of articles.

Carlos A. Lopez-Sanchez conceived and designed the experiments, performed the experiments, analyzed the data, contributed reagents/materials/analysis tools, prepared figures and/or tables, authored or reviewed drafts of the paper, approved the final draft, calculation data.

Sarahi Sandoval analyzed the data, contributed reagents/materials/analysis tools, prepared figures and/or tables, authored or reviewed drafts of the paper, approved the final draft, review of articles.

Marco A. Marquez-Linares analyzed the data, contributed reagents/materials/analysis tools, prepared figures and/or tables, authored or reviewed drafts of the paper, approved the final draft, review of manuscript.

Christian Wehenkel conceived and designed the experiments, performed the experiments, analyzed the data, contributed reagents/materials/analysis tools, prepared figures and/or tables, authored or reviewed drafts of the paper, approved the final draft, calculation of data and review of articles.

The following information was supplied regarding data availability:

The raw data are provided as Supplemental Dataset Files.

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
