# Peer review of "Predicting Pinus monophylla forest cover in the Baja California Desert by remote sensing"

_PeerJ, doi:10.7717/peerj.4603_

## Round 0.1 · original submission · Major Revisions

· Academic Editor

Major Revisions

Both reviewers agree on the potential importance of your contribution, but also recommend a substantial rework of your manuscript. Especially, the methodology and data analysis and interpretation needed to be addressed more carefully.

Reviewer 1 ·

Basic reporting

All issues are satisfactory.

Experimental design

The sampling design for the reference data is not reported. Failure to use a probability sampling design invalidates all the analyses.

The prediction/classification techniques must be described in much greater detail and some justification with references for data characteristic of this study must be included.

The apparent “pixel counting” method for estimating areas is a biased procedure because it does not account for classification error. Uncertainties associated with the area estimates should be included.

Validity of the findings

Until the methods are described in more detail, the sampling design for the reference data is provided, and an unbiased procedure for estimating area is used, the validity of all results are questionable.

Additional comments

GENERAL COMMENT:
The overall topic of the paper is timely and relevant, and a well-written paper using statistically rigorous methods would make an appropriate contribution to the literature. However, the paper in its current form lacks rigor and suffers from considerable incorrect and inappropriate use of terminology.

Because remotely sensed spectral data measure only the intensity of reflected light at different frequencies, these data do not “detect” or “establish” any ground conditions. However, variables based on these intensities have been shown to be good PREDICTORS of ground conditions. This is not just a subtle, semantic issue, because terms such as “detect” and “establish” fail to connote the uncertainty associated with the predictions. Further, if there were no such uncertainty, then assessment of accuracies based on confusion matrices would not be necessary. Lines 22, 26, 39, 91, 06, 100, 104, 198, 242. In nearly all these cases, either “predict” or “estimate” would be more correct terms.

All of the prediction and/or classification techniques must be described in greater detail and supported with appropriate references.

The authors must describe their methods for assessing areas. In particular, constructing a map and adding the areas of pixels classified into a particular class, a practice characterized as pixel counting, is biased because it does not take into account classification error. See Olofsson et al, 2013, Remote Sensing of Environment 129: 122-131. In particular, what is the uncertainty associated with the authors’ area estimates? Lines 37, 196.

The “real values” to which the authors refer at line 164 are more generally characterized as “reference data.” What sampling design was used to acquire these data? In particular, if a probability sampling design was not used, then all the results based on the confusion matrix analyses are incorrect. See the previous Olofsson et al, 2013 reference.

The first dictionary definitions of terms such as “high”, “low”, “over”, “under”, etc, refer to height, altitude, or vertical position, whereas the first dictionary definitions of terms such as “large”, “small”, “greater”, “less”, etc, refer to size or amount. Despite widespread use to the contrary in the applied literature, perhaps the latter terms can be used in scientific papers to refer to size or amount. In particular, if terms such as “higher” or “lower” are used to refer to altitude, they should not also be used to refer to amounts or spatial resolution of imagery. Lines 95, 104, 145, 200, 202, 224, 266, 282,

Avoid the use of subjective terms such as “best” whose criteria are not given. Lines 218, 221,

SPECIFIC COMMENTS:
Line 37: What is meant by “probably”?

Line 73: What is meant by “generalized”?

Line 104: Change “higher” to “finer.”

Lines 150, 198: Clarify these statements. For someone unfamiliar with the technique, these statements are meaningless.

Line 169: In a paper with statistical content, the term “significant” should be reserved for reporting the results of a formal statistical test of significance in which case both the name of the test and either the P-value or the alpha-level should also be reported.

Lines 183, 185: What is a “significant variable”? Is it that the mean of the variable is statistically significantly different from 0?

Line 196: What is meant by “potential”?

Line 228: This is a very small R2.

Line 235: What is meant by “efficient” and what is the evidence to support the assertion?

Line 278: How can temperature be “wide” or “narrow”?

Reviewer 2 ·

Basic reporting

Please see attached PDF.

Experimental design

Please see attached PDF.

Validity of the findings

Please see attached PDF.

Additional comments

Please see attached PDF.

Annotated reviews are not available for download in order to protect the identity of reviewers who chose to remain anonymous.

---

## Round 0.2 · Minor Revisions

· Academic Editor

Minor Revisions

Reviewer 1 stated that you should use stratified rather that simple random estimators for your data sets. Please revise this point in your data analysis.

Both reviewers note grammar/ wording issues. Please make sure that your manuscript is diligently corrected for language.

Reviewer 1 ·

Basic reporting

No comment

Experimental design

The stratified random sampling design is appropriate, but stratified rather that simple random estimators must be used with the acquired sample data.

Validity of the findings

The accuracies and the area estimates are not correct.

Additional comments

GENERAL COMMENTS:
The paper is much improved, although several wording revisions are necessary and one technical error must be corrected.

The technical error is that the authors used stratified random sampling to acquire their reference data, but they used simple random sampling estimators to assess the confusion matrix data. For each map class used as a stratum, the proportion of the study area in stratum must be calculated as the proportion of pixels assigned to the class/stratum and used as a stratum weight. The authors should carefully read and follow the examples in the Olofsson papers already cited.

Figures 4, 5, and 6 are not very meaningful when all the response variables are 0 and 1. An alternative is to order/sort all the data by the predicted probability. Then aggregate the pairs of observations and predictions into groups of a selected size, perhaps 15-20. Calculate the mean of the observations and the mean of the predictions in each group and graph mean observations versus mean predictions. If the predictions are good, the graph should fall along the 1:1 line. Although the study is quite different, see Figure 2 in Remote Sensing of Environment 114: 1017-1025.

The first dictionary definitions of terms such as “high”, “low”, “over”, “under”, etc, refer to height, altitude, or vertical position, whereas the first dictionary definitions of terms such as “large”, “small”, “greater”, “less”, etc, refer to size or amount. Despite widespread use to the contrary in the applied literature, perhaps the latter terms can be used in scientific papers to refer to size or amount. Technical writing consultants strongly recommend using terms with only one meaning or definition, or if a term with multiple definitions is necessary, then the use the first definition. A recently submitted paper used the terms “high resolution”, “high biomass”, and “high altitude” all in the same sentence. That might be extremely confusing for a reader with limited English language expertise. Lines 27, 235, 260, 292, elsewhere. This comment was provided for the previous version of the paper.

When uncertainty is involved, used terms such as “predict” or “estimate” rather than “determined” (139, 191, 208, 306), “obtained” (140), “identified” (163), “calculated” (194, 199), “measured” (212), elsewhere.

SPECIFIC COMMENTS:
Line 29: Change “proved” to something like “demonstrated” because rarely is anything actually proven in science.

Line 79: Perhaps “three times finer resolution.”

Line 145: Is “indices” the plural of “index”?

Line 173: Should “data” be “predictions.”

Line 292: Perhaps, “very large accuracies.”

Reviewer 2 ·

Basic reporting

Please see attached document.

Experimental design

Please see attached document.

Validity of the findings

Please see attached document.

Additional comments

Please see attached document.

Annotated reviews are not available for download in order to protect the identity of reviewers who chose to remain anonymous.

---

## Round 0.3 · Minor Revisions

· Academic Editor

Minor Revisions

Please address the comments of reviewer 1 concerning the reporting of the stratified estimators analysis.

Reviewer 1 ·

Basic reporting

Okay.

Experimental design

The stratified estimators of overall accuracy, areas, and standard errors must be reported in sufficient detail that the reader understand how they were calculated and can replicate the analyses. This is a non-trivial issue and requires that the revised paper be reviewed before acceptance.

Validity of the findings

Okay,

Additional comments

GENERAL COMMENTS:
Do not submit papers without both line AND page numbers.

The authors have used the stratified estimators correctly, but they need to include the estimators (the formulae) so the reader understands how the estimates were calculated. Readers should not have to search the literature for information that is crucial for understanding the current study. How was overall accuracy estimated? How were the area estimate and its standard error calculated? What is W_i in Table 4?

As per previous versions of this paper, the first dictionary definitions of terms such as “high”, “low”, “over”, “under”, etc, refer to height, altitude, or vertical position, whereas the first dictionary definitions of terms such as “large”, “small”, “greater”, “less”, etc, refer to size or amount. Despite widespread use to the contrary in the applied literature, perhaps the latter terms can be used in scientific papers to refer to size or amount. Technical writing consultants strongly recommend using terms with only one meaning or definition, or if a term with multiple definitions is necessary, then the use the first definition. Further, use of “high” to mean “large” is not well-defined. What are the rules for this usage? For example, would you characterize a large stone as “high”? Lines 44, 242, elsewhere. The authors stated in response to the last round of review that they had corrected this problem.

In statistics, the term “multivariate” refers to multiple dependent or response variables. What were the multiple response variables? Do not confuse “multivariate regression” meaning multiple dependent variables with multiple linear regression meaning multiple independent variables. Further, the term “multiple” is only used for linear regression whereas here it is logistic (not logistical) regression. Lines 261, 274, Table 4 caption, elsewhere.

SPECIFIC COMMENTS:
Line 63: How can a tree be distributed? Is it not the species that is distributed?

Line 37: Should “valuation” be “validation”?

Lines 40, 235: What does the number to the right of +/1 sign indicate? Standard error? Half-width of the confidence interval?

Line 77: Correct the typographical error: “the The.”

Line 77: First, the sensors are not used; rather it is the band values. Second, band values cannot be used to “analyze vegetation” because they only characterize the intensity of reflected light. Rather, they can be used to “predict” vegetation attributes.

Line 167: At best, the classification can be used to “predict” types of land cover, and then only with error or uncertainty.

For Table 3 (error matrix), place “Reference” as a horizontal heading across the top.

Reviewer 2 ·

Basic reporting

No more comments.

Experimental design

No more comments.

Validity of the findings

No more comments.

Additional comments

Thank you for making the necessary corrections. I will now recommend to the editor that your manuscript be accepted as it is. Cheers!

---

## Round 0.4 · accepted · Accept

· Academic Editor

Accept

In my opinion the comments of the reviewers have been addressed adequately.

#